# Ionizing Radiation-Induced Brain Cell Aging and the Potential Underlying Molecular Mechanisms

**DOI:** 10.3390/cells10123570

**Published:** 2021-12-17

**Authors:** Qin-Qi Wang, Gang Yin, Jiang-Rong Huang, Shi-Jun Xi, Feng Qian, Rui-Xue Lee, Xiao-Chun Peng, Feng-Ru Tang

**Affiliations:** 1Laboratory of Oncology, Center for Molecular Medicine, Health Science Center, School of Basic Medicine, Yangtze University, Jingzhou 434023, China; 201971481@yangtzeu.edu.cn (Q.-Q.W.); 201971484@yangtzeu.edu.cn (S.-J.X.); 2Health Science Center, Department of Pathophysiology, School of Basic Medicine, Yangtze University, Jingzhou 434023, China; 3Department of Neurology, Jingzhou Central Hospital, Jingzhou 434023, China; 201971487@yangtzeu.edu.cn; 4Health Science Center, Department of Integrative Medicine, School of Health Sciences, Yangtze University, Jingzhou 434023, China; Hjr@yangtzeu.edu.cn; 5Health Science Center, Department of Physiology, School of Basic Medicine, Yangtze University, Jingzhou 434023, China; qianfeng@yangtzeu.edu.cn; 6Radiation Physiology Laboratory, Singapore Nuclear Research and Safety Initiative, National University of Singapore, Singapore 138602, Singapore; snrlrx@nus.edu.sg

**Keywords:** ionizing radiation, aging, brain, oxidative stress, mitochondrial dysfunction, DNA damage

## Abstract

Population aging is occurring rapidly worldwide, challenging the global economy and healthcare services. Brain aging is a significant contributor to various age-related neurological and neuropsychological disorders, including Alzheimer’s disease and Parkinson’s disease. Several extrinsic factors, such as exposure to ionizing radiation, can accelerate senescence. Multiple human and animal studies have reported that exposure to ionizing radiation can have varied effects on organ aging and lead to the prolongation or shortening of life span depending on the radiation dose or dose rate. This paper reviews the effects of radiation on the aging of different types of brain cells, including neurons, microglia, astrocytes, and cerebral endothelial cells. Further, the relevant molecular mechanisms are discussed. Overall, this review highlights how radiation-induced senescence in different cell types may lead to brain aging, which could result in the development of various neurological and neuropsychological disorders. Therefore, treatment targeting radiation-induced oxidative stress and neuroinflammation may prevent radiation-induced brain aging and the neurological and neuropsychological disorders it may cause.

## 1. Introduction

Global population aging is currently occurring at an unprecedented rate. There has been a demographic shift toward an older population, and this may have far-reaching consequences. Population aging is considered a crisis from a global economy and healthcare perspective [1]. In most species, the geriatric stage of life involves the impairment of adaptation and self-balancing mechanisms, leading to increased susceptibility to environmental or internal pressure, disease, and mortality [2]. In humans, aging is associated with progressive cognitive and physical impairment, as well as an increasing risk of diseases such as neurodegenerative diseases. Age-related disability and morbidity negatively affect the quality of human life, ultimately increasing the risk of mortality, leading to problems at the individual, family, and community levels [3].

Brain aging, which involves complex cellular and molecular mechanisms that ultimately lead to cognitive decline, is the primary contributor to neurodegeneration [4]. Aging causes a gradual deterioration in the brain’s functional capacity, which leads to impaired learning and memory, attention deficits, reduced decision-making speed, and impaired sensory and motor incoordination [5]. The age-related deterioration of brain function occurs almost parallel to the functional deterioration of other organ systems, and the decline in performance is significantly accelerated after the age of 50 years [6]. Nonetheless, aging-related alterations in cellular integrity and molecular pathways are shared across tissues, including the brain [7]. These alterations include mitochondrial dysfunction; intracellular accumulation of oxidative damage to macromolecules; dysregulation of energy metabolism; impairments in cellular waste disposal (autophagy-lysosome and proteasome functions), adaptive stress response signals, and DNA repair; and inflammation. Further, abnormal neuronal network activity, altered Ca^2+^ processing in neurons, and reduced neurogenesis are also observed in the aging brain [8,9].

All living organisms undergo aging and are exposed to ionizing radiation (IR) throughout their lifespan. Several studies have linked IR to accelerated aging [10]. Kuzmic et al. used glp-1 sterile *Caenorhabditis elegans* to evaluate the impact of chronic gamma radiation on lifespan and confirmed that IR can accelerate aging [11]. Exposure to IR is known to cause a wide array of physiological changes. IR can lead to DNA double-strand breaks (DSBs), which cause genetic instability DNA damage and oxidative stress, leading to brain endothelial cell senescence and cell death [12,13]. Cellular senescence, an irreversible state of growth stagnation, can help us understand the relevance of aging to several other biological processes, from embryonic development to tissue repair and aging-related diseases [14,15]. High-dose exposure may cause acute radiation sickness, whereas prolonged exposure to low-dose radiation often results in chronic disorders such as neurodegenerative diseases. While the harmful effects of high-dose/dose-rate IR on human health are well-established, the effect of low-dose/dose-rate exposure is often overlooked despite its ubiquitous nature. The use of IR in medical diagnosis and cancer treatment has increased significantly. Consequently, nuclear waste from hospitals accounts for approximately 14% of the world’s total annual radiation exposure [16]. Several studies have shown that long-term exposure to low-dose IR in catheterization laboratories increases the risk of cardiovascular diseases, indicating that it causes enhanced vascular aging and early atherosclerosis [17]. More than 50% of cancer patients will be treated with radiotherapy. Radiotherapy will kill the tumor tissue while also damaging the surrounding normal tissues, leading to radiation toxicity. [18]. Radiotherapy exposes both tumor tissue and surrounding healthy IR, causing DNA damage, which triggers the DNA damage response (DDR). In this reaction, ionizing radiation will cause the cell cycle to stop and cause cell damage, and then these damaged cells will automatically repair. If the DNA is fully repaired, these cells can recover as before. However, when internal and external factors affect the ability of DNA repair, senescence (i.e., permanent cell cycle arrest) or cell death (such as apoptosis or mitotic disaster) will occur. [19,20]. Furthermore, as IR is often required for obtaining high-resolution images during neuroimaging, the contribution of low-dose/dose-rate IR toward neurodegenerative diseases must be examined [21].

Radiation exposure, particularly natural radiation exposure, occurs in daily human life. Some radioactive elements in the earth’s crust such as uranium (238U), potassium (40K), thorium (232T), and their radioactive decay products, e.g., radon (222Rn) and radium (226Ra) act as natural sources of radiation exposure [22]. Areas with high levels of background radiation are considered ideal for investigating the long-term effects of chronic low-dose radiation exposure in humans [23]. Some studies on high natural background radiation have been performed in Brazil, China, India, and Iran [24,25,26]. High-dose radiation exposure can cause cancer. In addition, it is worth noting that the high natural background radiation observed in Yangjiang, China also increases the incidence of some non-cancer diseases, such as tuberculosis, digestive diseases and cerebrovascular diseases. [27]. High natural background radiation in the environment can be considered as a type of natural pollution. It can reach the human body through both internal and external sources, and it can damage human DNA. In addition, natural background radiation also enters the ecosystem through human activities, affecting the health and quality of life of individuals residing in areas with high natural background radiation.

A long-term follow-up study of the 1986 Chernobyl disaster revealed an increased incidence of an extensive array of diseases in exposed individuals across all contaminated regions assessed. In particular, alterations to the central nervous system (CNS), resulting in radiation-induced neurocognitive dysfunction, were observed in many individuals [28]. Further, surviving Chernobyl liquidators showed signs of inflammation that could be associated with premature aging [29]. Radiation-induced immune system impairments are important contributors to the physiological changes that occur shortly after radiation exposure and have been implicated in delayed effects of radiation such as tumor development and early aging [30]. Chronic low-dose IR exposure can accelerate the aging of blood vessels, including cerebral vessels. This has been shown to correlate with age-related encephalopathy in individuals over 40 years of age, as well as with systemic atherosclerosis [31,32]. Of the 306 workers exposed to the Chernobyl nuclear accident examined in a previous study, 81% and 77% of men and women, respectively, exhibited signs of accelerated aging. In addition, those younger than 45 years of age appeared to be more susceptible to radiation-induced accelerated aging [33]. In humans, sensitivity to radiation decreases with age until an individual matures. However, this sensitivity increases in old age.

Experimental data from animals also supports the theory that IR induces aging. Brizzee observed that with increasing age, some changes occur in the cerebral cortices of Rhesus monkeys and albino rats [34]. Analyses of transcriptomic profiles from murine brains revealed that the molecular responses observed hours after full-body low-dose irradiation (100 mGy) were similar to those associated with premature cognitive decline, Alzheimer’s disease, and various neuropsychiatric disorders [35,36]. The transcriptomic profiles of microglia obtained one day and one month post-irradiation were also similar to those observed during aging, pointing to the aging-enhancement effects of radiation [37]. In vitro high-dose (2–8 Gy) irradiation of primary cerebrovascular endothelial cells in rats promotes a secretory phenotype associated with aging, characterized by the upregulation of pro-inflammatory cytokines and chemokines, including IL-6, IL-1α, and MCP-1 [38]. It has been reported that IR increases cellular senescence, and senescence-associated β-galactosidase (SA-β-Gal) and senescence specific genes (p16, p12, and Bcl-2) are highly expressed in irradiated bone marrow derived macrophages [39]. These findings corroborate the in vivo evidence pointing to the potential senescence-inducing effects of radiation on the endothelial cells of cerebral blood vessels.

Altogether, it is clear that the biological effects of IR exposure, not limited to oxidative stress, chromosomal damage, apoptosis, stem-cell failure, and inflammation, all contribute to accelerated aging [40]. Furthermore, the contribution of IR exposure to the development of non-malignant conditions such as neurodegenerative diseases is also becoming evident through epidemiological studies [21], and many medical conditions have been found to be related to exposure to different types of low-dose/dose rate radiation (Table 1).

In this review, we will focus on the impact of IR on brain aging, including the aging of various CNS cell types (microglia, astrocytes, cerebral endothelial cells, and neurons). Further, the relevant molecular mechanisms will be discussed, and future research directions aimed at elucidating the true impact of radiation-induced brain aging will be proposed.

## 2. Radiation-Induced Senescence of Different Types of Brain Cells

The understanding of how IR affects brain aging begins with an elucidation of its influence on individual CNS cell types. CNS cells are broadly classified into two categories, glial cells—including microglia, astrocytes, and oligodendrocytes—and neurons [45]. The endothelial cells of cerebral blood vessels are also closely associated with the brain; they lead to the formation of the blood–brain barrier (BBB) and are important for maintaining CNS integrity [46]. IR has been proven to cause to happen aging in all these cell types, especially microglia, astrocytes, cerebral vascular endothelial cells, and neurons (Table 2). The cumulative effects of senescence in these cells may lead to brain aging, related neurological and neuropsychological disorders, and a shortened lifespan [40].

### 2.1. Microglia

Microglia are yolk sac-derived phagocytes located in the CNS [51]. These cells are involved in immune responses and the maintenance of brain homeostasis. Microglia also respond to changes in the tissue environment by upregulating different cell surface receptors and producing a multitude of secreted factors [52]. As such, this class of glial cells has been affected in a glut of neurological diseases. Under normal physiological conditions, there is a balance between pro- and anti-inflammatory mediators in the brain [53]. However, a shift towards the pro-inflammatory state is observed during brain aging. Interestingly, this shift is also observed after radiation exposure [54]. Thus, radiation-induced neuroinflammation could be a potential contributor to the development of brain aging and cognitive impairment [55].

Although microglial activation is necessary for protection against foreign substances, beyond a certain threshold, such activation can be damaging. Activated microglia exhibit a neurotoxic phenotype and can cause neuronal damage and death. This phenotype has been implicated in various neurodegenerative diseases, radiation-induced brain injury, and brain aging [47,56]. Several features of this neurotoxic phenotype resemble those of aging. In microglia from aged mice, pro-inflammatory cytokines such as IL-6, IL-1β, and tumor necrosis factor-α (TNF-α) are upregulated [57], and telomeres appear to be shortened [58]. These factors are also upregulated in murine models of accelerated senescence [59]. In addition, microglia abnormally activated by radiation continuously produce neurotoxic cytokines, including IL-1β, TNFα, and IL-6 [60]. Moreover, these cells show elevated reactive oxygen species (ROS) levels, inducing oxidative stress and consequent DNA damage [61]. Furthermore, in vitro, chronically activated microglia exhibit multiple features of aging, leading to SA-β-Gal activity, metachromatic focus formation, and growth arrest [62].

Chronic aging is considered to be the driving force for age-related tissue dysfunction. Studies show that SA-β-Gal and p16^INK4a^, key markers of senescence, are up-regulated in microglia treated with irradiation. Additionally, these markers continue to be expressed even one month post-irradiation. Senescent cells often secrete large amounts of cytokines and matrix metalloproteinases (MMPs) and show sustained oxidative and genotoxic damage, eventually leading to tissue impairments and aging [63].

IR leads to progressive DNA damage; moreover, the consequent induction of oxidative stress can trigger the aging of normal cells [64]. Studies have demonstrated that IR can induce senescence in microglia [65], characterized by inflammation, DDR, and metabolic changes [66]. Irradiated microglia are involved in the pathology of radiation-induced brain damage [56] and aging-related diseases [48].

### 2.2. Astrocytes

Astrocytes, one of the most common brain cells, were previously considered non-functional cells providing packing for brain networks. Nevertheless, recent research has demonstrated their functional roles in several processes. Previous studies have reported that microglia and astrocytes interact with each other, and astrocytes are known to participate in immune activity [67]. Astrocytes provide osmotic balance and therefore contribute to the maintenance of CNS homeostasis [68]. They also provide metabolic support to neurons [69] and help in the establishment and maintenance of the BBB [70]. Astrocytes promote neuronal communication and are involved in neurotransmitter recovery. They also help protect the brain against trauma, infections, and neurodegeneration, thus maintaining its health and function [71].

With age, the number of astrocytes expressing p16^INK4a^ and MMP3 (protease closely associated with the senescence-associated secretion phenotype) increases [72]. Primary astrocytes isolated from human brain tissues acquire senescence-related characteristics after multiple passages [48]. Incidentally, exposure to IR has also been found to induce the senescence-associated secretory phenotype (SASP) and aging in irradiated human astrocytes, likely due to excessive DNA damage accumulation [73]. In a mouse model of radiation-induced brain injury, changes in the expression of TNF-α and IL-1β mRNA and related signaling pathways have been observed in the hippocampus. It has also been observed that the release of pro-inflammatory cytokines and inhibition of hippocampal neurogenesis may be related to the activation of microglia and may play a critical role in radiation-induced brain injury [74]. Additionally, irradiated astrocytes showed an increase in the expression of the senescence-related markers p16^INK4a^ and p21, cell size, and number of multinucleated cells [75]. In contrast, they showed a decrease in cell number and the expression of glial fibrillary acidic protein (GFAP), which is also observed during aging [76]. Altogether, these findings suggest that irradiated astrocytes not only promote neuroinflammation but also contribute to radiation-induced accelerations in brain aging.

### 2.3. Brain Endothelial Cells

Radiation-induced senescence is also observed in brain microvascular endothelial cells [49]. Cell surface proteins on brain endothelial cells communicate with both the blood and brain. Thus, these cells are involved in signal transmission and transduction across the BBB. Exposure to IR can cause premature degeneration of endothelial cells and thinning of the cerebral blood vessels, resulting in the temporary loss of contextual learning, interruption of working memory, gradual spatial learning impairments, and an increased risk of dementia [77].

Both cell culture and live animal studies have shown that radiation promotes stress-induced progeria-like phenotypes in endothelial cells [78,79,80]. Studies of brain microvascular endothelial cells from rats exposed to γ-irradiation have demonstrated that cerebral vascular endothelial cells are more radiosensitive than microglia and neurons. γ-irradiation has also been observed to destroy the clone formation and proliferative abilities of cerebral vascular endothelial cells, which are essential for intracranial angiogenesis. Cerebral microvascular damage caused by γ-irradiation has also been found to promote the accelerated aging of healthy tissues and result in progressive cognitive decline in <50% of tumor patients receiving radiotherapy [81].

In addition, acute γ-irradiation leads to increased production of ROS in cerebral vascular endothelial cells, causing premature aging [82]. Concomitantly, IR may also induce the expression of p16^INK4a^, the main driver of cell cycle arrest during cerebral vascular endothelial cell senescence, leading to permanent cell cycle arrest [83,84]. Such radiation also increases the proportion of SA-β-Gal-positive cerebral vascular endothelial cells [3]. Similar to astrocytes, cerebral vascular endothelial cells also express the IR-induced SASP, including an enhance in the production of pro-inflammatory molecules, cytokines, chemokines, growth factors, and MMPs [85]. Therefore, radiation-induced SASP may promote neuroinflammation and cause neuronal damage by altering the microenvironment of endothelial cells [86]. Recently, Remes et al. examined the epidemiology of cerebrovascular disease in long-term childhood brain tumor survivors 20 years after the end of radiotherapy. They found that the incidence of ischemic infarction, microhemorrhage, and lacunar infarction in this population was similar to or higher than that observed in the general population over 70 years of age [87]. Such clinical data supports the hypothesis that radiation triggers the accelerated aging of the cerebrovascular system.

### 2.4. Neurons

Neurons are the most basic structural and functional unit of the nervous system. They interact with other functional cells in the brain and influence each other. Microglial senescence has a profound effect on neuronal activity and cognition during natural aging [88]. In a study on the effects of age on the response to radiation, it was observed that although the number of immature neurons in old rats did not decrease continuously after whole-brain irradiation, the inflammatory reaction was greater than that in younger rats. Thus, this reaction may have a greater contribution to the development of radiation-induced cognitive impairments in older adults [50]. 

Furthermore, neurons are one of the most highly oxygenated cells and experience oxidative genomic damage after long-term exposure to endogenous ROS, a by-product of cellular respiration [88]. These cells are extremely vulnerable to the DNA damage induced by genotoxic substances such as oxidative stress and IR. When DNA damage occurs, powerful DNA repair mechanisms are activated to limit the accumulation of oxidative damage [89]. However, the accumulation of unrepaired DNA may cause aging and several neurodegenerative diseases [90]. Chronic low-dose-rate γ-irradiation can induce brain aging and reduce neuronal density [34]. In mouse studies, high-dose rate γ-irradiation was shown to reduce the activity of superoxide dismutase and increase the amounts of free radicals, which may be related to aging [91].

In mice, several hours after whole-body irradiation (100 mGy), expression-level changes in molecules and networks involved in cognitive function, advanced aging, Alzheimer’s disease, and neuropsychiatric diseases are observed [35,36]. Tang et al. studied the expression of γH2AX in the brains of mice exposed to radiation on different postnatal days. They suggested that persistent radiation-induced DNA damage at 120 days and 15 months after irradiation in the early life of mice may be associated with brain aging and shortened life expectancy [92]. Radiation-induced oxidative stress and inflammation prevent neurogenesis in the subgranular zone and induce aging of the granule cell assembly, leading to cognitive impairment [93,94,95].

## 3. Effect of Radiation-Induced Brain Aging

With an increase in life expectancy owing to improvements in medical interventions, it has become crucial to understand the advantages of radiation protection among older individuals. Increased inflammation, loss of the redox balance, continued telomere wearing, decreased efficiency of the DDR, mitochondrial dysfunction and autophagy are all changes that occur during aging and can negatively impact genome integrity (Figure 1). Further, as radiation can exacerbate these changes, it is important to understand the mechanisms involved in the effects of radiation exposure and brain aging. Radiation-induced ROS can directly cause mitochondrial respiratory chain breakage and water molecular decomposition inducing respiratory chain dysfunction and reduced antioxidant capacity. NADPH oxidase is a family of multi-subunit complex enzymes that activate the conversion of oxygen to superoxide anions (O_2_^−^) with NADPH as the electron source, and exists in vascular endothelial cells. In addition, cyclooxygenases-2 (COX-2) and 5-lipoxygenase (5-LPO) catalyze the production of prostaglandin H2 (PGH2) and the formation of ROS during arachidonic acid metabolism. 

Cells exposed to IR can be destroyed directly by secondary electrons and/or indirectly by ROS, resulting in DSBs [18] followed by the DDR [19,20,96,97,98,99,100]. The onset of this response generally does not exceed a few minutes after DNA damage. The damaged DNA is recruited by a complex reaction network that depends on damaged cells. The response produced by the cell depends on the type of DNA damage and environmental factors. For example, a response through transient activation of cell cycle checkpoints and DNA repair, namely cell survival. [101]. DSBs are usually repaired by error-prone non-homologous terminal junctions, which are coordinated by DNA-dependent protein kinases. Homologous recombination is operated only during the S or G2 phases of the cell cycle [101]. Unrepaired and/or incorrectly repaired DSBs may lead to genomic instability and cell death or cell senescence (an irreversible state of cell cycle arrest) [19,102]. Radiation exposure directly alters mitochondrial DNA, most notably the common deletion mutation. IR also indirectly alters mitochondrial dysfunction by producing ROS, resulting in disruption of the electron transport chain, and increases the production of antioxidant enzymes through nuclear factor E2-related factor 2 (Nrf2) [103].

Cells enter senescence with shortened telomeres, and, in fact, short telomeres increase the sensitivity of cells to radiation, with human cells that are sensitive to radiation having shorter telomeres than normal cells [104]. Individuals with short telomeres have a higher frequency of radiation damage than individuals with long telomeres [105]. The mechanism of telomere maintenance is directly or indirectly related to DNA damage [106]. Telomere shortening is an important sign of aging.

Under normal conditions, nuclear factor κB (NF-κB) associates with the inhibitory protein (IKB) to compose a protein complex and remains sleepy. However, when cells are exposed to IR, NADPH oxidase activity and other oxidative stress reactions are increased, mitochondrial electron transfer is impaired and occurs rapidly. ROS are over-generated in oxidative stress reactions, and IL-1 and TNF are produced by inflammatory cells to link to IL-1 receptor and TNF receptor, respectively, which in turn activates downstream NF-κB signaling and leads to transcription of inflammation-related genes. Activated NF-κB induces COX-2 and 5-LPO expression, leading to ROS production, which forms a positive feedback loop to increase inflammation and oxidative stress. Intracellular ROS directly stimulate NF-κB which may up-regulate the expression of cytokines including IL-1 and TNF. These cytokines increase inflammation by attracting white blood cells and activating NF-κB.

The way autophagy maintains protein production includes promoting misfolding and degradation of aggregated proteins. Mitochondrial quality control involves removing damaged mitochondria through autophagy. Autophagy cargo receptors recognize ubiquitin-modified mitochondria, facilitating their sequestration within autophagosomes and lysosome-mediated breakdown. The molecular pattern related to mitochondrial damage is also one of the reasons for triggering the production of inflammatory cytokines. The method of autophagy to regulate senescence is to promote the disintegration and degradation of the nuclear lamina. Free amino acids released from the lysosome during senescence support anabolic activities, including the production of inflammatory cytokines that make up the SASP (Figure 2).

### 3.1. Oxidative Stress

Endogenous ROS production is a byproduct of regular cell metabolism, but exogenous ROS production may also occur due to radiation and chemical compounds [107]. ROS can be produced from a number of sources after irradiation [108]. Classical radiobiology shows that compared with most other oxidative stresses, the amount of ROS produced by the radiolysis of water is smaller and the maintenance time is shorter. Exposure to ionizing radiation will generate ROS in the 2 nm range of DNA and form complex DSBs, thereby triggering high cytotoxicity. The mitochondrial membrane-bound nicotinamide adenine dinucleotide phosphate oxidases (NOX) or other oxidases may have an indirect connection with the DDR pathway. These oxidases are the main source of cellular ROS caused by oxidative stress. [109,110,111]. Radiation can also cause ROS generation from these sources by damaging the mitochondria, stimulating NOX or other oxidases [110,111], leading to ATP release, ion channel activation [112] and purinergic signaling [113]. One example of a damage-associated molecular pattern (DAMP) observed after irradiation is expression of the high-mobility group box 1 (HMGB1) protein, a chromatin binding nuclear protein that acts via toll-like receptor 4 (TLR4) signaling to promote further ROS production [114]. High ROS activity can directly destroy macromolecules such as lipids, nucleic acids, and proteins. DNA damage, typically in the form of strand breaks and cross-links, leads to genomic mutations. Furthermore, ROS cause high oxidative stress in affected cells. In cells, oxidative damage depends on ROS concentration and on the balance between the relative levels of ROS and antioxidants. When the oxidant-antioxidant balance is lost, oxidative stress occurs, altering and destroying several intracellular macromolecules, as mentioned previously [115].

The free-radical theory of aging was put forth by Harman in 1956, and he subsequently demonstrated that mitochondrial respiration is the primary endogenous source of oxidative stress [116]. Aging is accompanied by increases in ROS levels and decreases in the activity and expression of antioxidant enzymes, including superoxide dismutase, catalase, and glutathione peroxidase [117,118]. Additionally, radiation-induced damage shows several characteristics often typical of cellular wear-and-tear, such as somatic mutations, which can trigger to the development of aging-related diseases [119]. ROS and reactive nitrogen species (RNS) attack macromolecules and cause oxidative stress, and this process has been implicated in several diseases. It has been also found that even low levels of ROS and RNS can lead to brain aging [120].

The inherent sensitivity of cells to radiation is thought to be dependent on the resultant ROS production. The irradiation of aging cells that already contain large amounts of active oxygen will undoubtedly overwhelm the antioxidant system responsible for removing excess amounts of oxygen metabolites [121]. The antioxidant protection against therapeutic and relieved doses of IR in human blood decreases with age [122]. These studies illustrate the role of oxidative stress regulation systems in determining the radiation sensitivity of senescent cells.

### 3.2. Mitochondrial Dysfunction

Mitochondria are present in neuronal dendrites and axons of neurons, and they produce the adenosine triphosphate (ATP) required for electrochemical neurotransmission and cell maintenance and repair [123]. As cells and organisms age, the efficiency of the electron transport chain (ETC) tends to decline, increasing electron leakage and reducing ATP production [124]. Most brain cells show the gradual accumulation of dysfunctional mitochondria, as observed in comparative studies of neurons and astrocytes in mice of different age groups [125,126]. With increasing age, mitochondrial dysfunction leads to increased ROS production, in turn causing further mitochondrial degradation and overall cell damage [127].

The main role of active oxygen is to activate the compensatory steady-state response. If ROS levels increase over a certain threshold, an imbalance in ROS homeostasis occurs, ultimately aggravating age-related damage [128]. Mitochondrial dysfunction can also cause aging via ROS-independent pathways. Mitochondrial defects may affect apoptosis signals by increasing the susceptibility of mitochondria to stress responses [129] and trigger the inflammatory response by promoting ROS-mediated and/or permeability-related inflammasome activation [124]. In addition, dysfunctional mitochondria could exert a direct influence on cell signaling and inter-organ crosstalk via negative effects on the outer mitochondrial membrane–endoplasmic reticulum interface [130].

Mitochondria isolated from animal brain tissue show many age-related changes, including increased mitochondrial DNA oxidative damage [131], mitochondrial enlargement or fragmentation [132], increased numbers of mitochondria with depolarizing membranes [133], and impaired ETC function [134]. Mutations and deletions of mitochondrial DNA (mtDNA) in older individuals may also lead to aging [135]. Due to the oxidative microenvironment in the mitochondria, mtDNA lacks protective histones. The efficiency of mtDNA repair mechanisms is also lower than that of nuclear DNA repair mechanisms. Hence, mtDNA is thought to be the main target of aging-related somatic mutations [136]. During brain development, abnormal mitochondrial breakage can lead to mitochondrial dysfunction and excessive ROS production, ultimately resulting in brain cell aging, cognitive impairment, and abnormal behavior [137].

IR exposure can induce mitochondrial dysfunction, which indirectly triggers aging. The time course of changes that occur after exposure to five Gy of γ-irradiation is as follows. First, cellular ROS levels increase significantly during the first few minutes, but reduce within 30 min. Subsequently, mitochondrial dysfunction is detected 12 h post-irradiation, as demonstrated by a decrease in the activity of nicotinamide adenine dinucleotide (NADH) dehydrogenase, the primary regulator of ROS release from the ETC [138].

Limoli et al. examined the mitochondrial membrane potential in unstable GM10115 cells after radiation exposure. They observed that the number of dysfunctional mitochondria was increased, and the mitochondrial membrane potential was decreased [139]. IR ionizes water molecules (H_2_O), mainly resulting in the production of •OH, the ROS with the highest damage-causing ability [140]. •OH can oxidize biological molecules, such as proteins and lipids [121,141]. The inner mitochondrial membrane contains phospholipids, such as phosphatidylcholine, phosphatidylethanolamine, and cardiolipin, which are required for optimization and aid the functions of various enzymes of the mitochondrial ETC [142,143]. Any change in the lipid profile of the membrane, such as a decrease in the lipid content and peroxidation, may lead to or enhance the production of O_2_^−^ via electron leakage from ETC enzymes. Therefore, the •OH produced due to irradiation induce mitochondrial oxidation via phospholipid peroxidation, thereby promoting O_2_^−^ production.

### 3.3. Telomere Attrition

Telomeres are special nuclear protein complexes that protect the ends of linear chromosomes in eukaryotic cells. Telomeres are bound by a characteristic polyprotein complex termed shelterin [144] which prevents the entry of DNA repair into telomeres which would otherwise be “repaired” owing to the presence of apparent DNA breaks, leading to low capacity for repairing DNA damage in this region. Therefore, telomere damage often induces cellular senescence and/or apoptosis [145,146]. The loss-of-function of shelterin components induces the rapid weakening of tissue regeneration and accelerates aging, even when telomeres are of a normal length [147].

IR can induce cell proliferation, apoptosis and senescence, all of which are associated with telomeres, through oxidative damage and DNA disruption. A study on peripheral blood obtained from 83 Chernobyl cleaners showed that compared with those in healthy blood donors, the relative length of telomeres in Chernobyl cleaners was significantly shorter. This study suggested that low-dose irradiation led to telomere shortening, and the alterations were sustained even 20 years post-irradiation [148]. Microglial senescence may also be related to the shortening of telomeres. The reduction in telomere length in microglia may lead to the ability of these cells to respond appropriately to CNS damage or infection, leading to apoptosis [149]. It is important to note that in the brain of patients with Alzheimer’s disease, malnutrition of microglia shows the strongest association with the degeneration of tau positive neurons, i.e., tau pathology is associated with microglial malnutrition. Thus, instead of brain inflammation, the lack of microglial support presents to be the essential cause of neurodegeneration. The age of microglia may be related to telomere shortening. Telomeres are the ends of eukaryotic chromosomes and shorten with age, leading to a sense of “replication” in microglia with self-renewal capabilities [150]. A study of 20 elderly patients with advanced head and neck cancer by Unryn et al. showed that radiation therapy resulted in severe shortening of telomere length in all patients [151]. Zhang et al. examined the effects of telomere dysfunction through the telomeric repeat binding factor 2 (TRF2)-mediated inhibition of neurons and mitotic nerve cells (astrocytes and neuroblastoma cells). They demonstrated that telomere dysfunction triggers DDRs and induces the activation of p53 and p21 and senescence [152]. Cellular responses to IR include cell cycle checkpoint arrest and programmed cell death. Since radiation outcomes double strand breaks in DNA leading to a reduction in telomere length, the radiation response appears to result from inappropriately induced cellular senescence [153].

The mechanism underlying accelerated aging due to IR is the same as that underlying ROS-mediated aging. Radiation damage to telomeres is also similar to the oxidative damage observed in these regions and leads to further shortening of telomeres and accelerated cell aging. A large dose of IR can cause tremendous cell death, resulting in compensatory cell division. Accelerated proliferation leads to telomere shortening, thereby accelerating aging in the entire organism, as observed in individuals exposed to IR [154]. Therefore, telomere shortening can be considered one of the mechanisms underlying radiation-induced aging [155].

### 3.4. DNA Damage

IR induces DNA damage through both direct and indirect pathways. The direct pathway refers to DNA ionization via radiation energy, and the indirect pathway refers to the generation of a large number of ROS after the radiolysis of water molecules. The latter pathway can induce DNA damage through a variety of mechanisms, including base damage and release, depolymerization, cross-linking, and chain breaking [156]. Such DNA damage, especially DSBs, triggers the complex and highly regulated DDR and repair pathways. DNA ionization directly causes damage to genetic macromolecules, whereas cytosol ionization leads to the generation of active substances such as •OH. Hydrated electrons and hydrogen atoms, which diffuse into the nanoscale area surrounding ionization events, inevitably react with the DNA components and indirectly damage them. Some of this damage occurs on the genetic molecular chain, leading to the generation of aggregated DNA damage or multiple damage sites. Owing to the endogenous nature of such damage, it is more likely to undergo error-prone repair than sparsely distributed damage, leading to irreversible cell damage [157].

At the molecular level, irradiated microglia show the upregulation of genes associated with DDRs, cellular stress, cell cycle arrest, and oxidative stress pathways [158,159]. Cells have a highly conserved and complex DNA damage recognition and repair network (DDR) that they use to respond to various types of DNA damage [160]. Studies using commercial cell lines and primary culture have shown that DNA damage can lead to permanent cell cycle arrest [161], resulting in an irreversible state in which damaged cells can survive but cannot proliferate, known as cellular senescence [162]. In mature neurons, homologous recombination and non-homologous terminal junctions are not sufficient to appropriately repair DSBs, likely because these cells do not divide. Therefore, it is generally believed that neurons increase unrepaired DNA damage over time, which may contribute to neurodegenerative diseases [163]. At acute stages after radiation exposure, radiation leads to p53-mediated speedy primary apoptosis and tardy secondary apoptosis (associated with mitotic mutations), thereby eliminating seriously damaged cells. Cells in the brain often respond to p53 activation through permanent cell cycle arrest instead of apoptosis [164].

With aging, DNA damage accumulates, inducing the loss of cellular function and the degeneration of cells and tissues. However, faulty repair can result in mutations and chromosomal aberrations. Unrepaired DNA damage often causes cell dysfunction or senescence, leading to multiple pathologies and cell death during aging [165,166]. The stimulation of DDR by IR triggers innate and adaptive immune regulation. Continued activation of the DDR promotes the production of inflammatory cytokines, involving IL-6 and IL-8 [167], initiating inflammatory responses that may damage surrounding tissues. Notably, DNA damage can occur not only in the nucleus but also in the mitochondria, and mtDNA is more vulnerable to damage than nuclear DNA. ROS is the primary source of mtDNA mutations, which can accumulate with age and disease progression [168,169].

Furthermore, the DDR leads to cell senescence, and continued senescence induces the SASP, in which inflammatory cytokines are released. These mediators influence neighboring cells and trigger various pathologies [170].

Unrepaired DDR induces cell senescence via the p53 pathway, activates the SASP, causes the secretion of pro-inflammatory cytokines, and further activates the innate immune response, resulting in tissue senescence and age-related diseases [171,172,173].

Senescent cells are characterized by increased activity of the most common senescence marker, SA-β-gal and the enlarged expression of p21^WAF1/Cip1^. While p21 can promote cellular senescence under exposure to IR, it can also promote G1 cell cycle arrest. However, this is not always correlated with p53 activity. Both p53 and p21 activation in senescent cells are temporary, since p53 and p21 expression decreases eventually and p16Ink4A maintains growth arrest in senescent cells [174]. In direct contrast to replicative senescence, stress-induced premature senescence (SIPS) caused by DNA damage is independent of telomere length or function [175,176].

### 3.5. Inflammation

Inflammation is a defensive response in the body. Chronic inflammation is linked to the onset and/or progression of a variety of diseases, such as age-related lesions and neurodegenerative diseases [177,178]. IR induces microglial activation and the release of inflammatory cytokines and chemokines. Inflammation is a common characteristic of microglial aging and plays a vital role in radiation-induced brain damage [179] and aging-related diseases [55]. As mentioned previously, radiation exposure often results in abnormal microglial activation, causing these cells to continuously produce neurotoxic cytokines, the levels of which increase with the dose of radiation [4]. DDR signaling can also be mediated by paracrine/systemic mechanisms that shape the systemic environment by regulating tissue repair and immune responses. Sustained DNA damage signals (telomere attrition) can cause DDR to send extracellular signals and induce SASPs [180,181,182]. The DDR/SASP signaling pathway regulates several bioactive pro-inflammatory mediators, such as interleukin-chemokine growth factor matrix degrading enzymes and ROS [183]. Moreover, the proinflammatory transcription of NF-kB and the inflammasome are the primary factors that set up the secretome, further highlighting the functional contribution of this pathway in the response to tissue injury [184,185,186]. NF-kB transcription triggers s to the production of several inflammatory features of SASP, such as IL-6, IL-1, and TNF-α, which are vital autonomic cellular modulators of aging [187,188]. Furthermore, a single dose of 10-Gy γ-irradiation can increase the levels of IL-6 and IL-8 in human endothelial cells in vitro [189]. In addition, an elevation in the inflammatory mediators TNF-α, IL-6, and IL-10 is observed with increasing radiation doses and age among survivors of atomic bombs [190].

Both IR and inflammation are related to an increase in ROS levels in tissues. In a mouse limb ischemia model, acute irradiation with two Gy was found to promote mast cell recruitment and tissue revascularization [191]. High-dose irradiation of the rat abdomen leads to neutrophil recruitment into the irradiated tissue [192]. Radiation-activated microglia express an inducible NO synthase and generate large amounts of NO, leading to neuronal oxidative damage. In addition, microglial toll-like receptors (TLRs) are involved in neuroinflammation, thereby contributing to age-related brain diseases [193]. Chronic inflammation may lead to excess ROS and RNS production, resulting in DNA damage and disease. The persistent presence of ROS and RNS in the microenvironment can lead to the further development of chronic inflammation, causing oxidative damage to DNA and DNA repair pathways, further leading to senescence and age-related diseases. In age-related neurodegenerative disease models, experimental activation of microglial TLRs can aggravate neuron degeneration, and pharmacological inhibition of microglial activation shows neuroprotective effects [194]. In addition to microglial activation, radiation-induced telomere shortening can also contribute to inflammation. Short telomeric ends induce DNA damage repair responses, leading to the production of NF-kB, a key regulator of inflammatory components such as the nod-like receptor 3 inflammasome and the secretion of inflammatory cytokines in the brain [195].

### 3.6. Autophagy

IR can cause macromolecular (mainly DNA) damage and endoplasmic reticulum (ER) stress induction, both of which can induce autophagy. [196]. Among the key molecules activated during radiation exposure, the inducible nitric oxide synthase (iNOS) gene and nitric oxide (NO) are involved in radiation induced autophagy and apoptosis [197,198]. The activation of the iNOS promoter will increase the production of NO, leading to the induction of autophagy mediated by protein nitration. The activation of iNOS promoter is related to its containing multiple transcription factor motifs such as NF-κB and kruppel like factor 6 (KLF6). [197]. Radiation-induced oxidative stress not only causes DNA damage, but also causes ER stress, impaired mitochondrial function, and protein misfolding. Most of these factors have been shown to induce autophagy [199,200].

Radiation-induced mitochondrial dysfunction and biogenesis are known to be related to mitochondrial autophagy [201]. Under conditions of extensive mitochondrial damage, the cell undergoes mitophagy so as to eliminate the damaged and dysfunctional mitochondria. Radiation induces a variety of responses, including autophagy and senescence. It is commonly thought that autophagy and senescence may promote cell survival. However, preclinical studies have demonstrated that autophagy can sometimes have opposite effects, such as cytotoxicity or other non-protective effects [202]. Free amino acids released by lysosomes during aging support the production of inflammatory cytokines that synthesize SASP [203].

Autophagy regulation is the core link of aging, age-related diseases and neurodegenerative diseases. In the process of aging and neurodegeneration, the regulation of autophagy will have step defects, leading to the accumulation of damaged organelles and protein aggregates, affecting cell metabolism and homeostasis, thereby exacerbating autophagy-related dysfunction and forming a vicious circle, which eventually leads to neuronal damage and cell death. [204]. Impaired autophagy in neurons contributes to the aggregation of toxic proteins and damaged organelles associated with neurodegenerative diseases [205]. Age is a vital risk factor for many neurodegenerative diseases, such as Alzheimer’s disease, Parkinson’s disease, and tauopathy [206]. Over time, the age-dependent decline in autophagy and the corresponding decline in protein metabolism and accumulation of protein toxicity together contribute to disease development and/or progression. Since post-mitotic neurons cannot eliminate protein-toxic damage in daughter cells during mitosis, they are more susceptible to age-related protein toxicity [207]. Impaired autophagy in glial cells, which have a critical homeostatic role in the central nervous system, may influence autophagic activities in neurons [208]. Altogether, these factors can interact and promote the degeneration of specific neurons in different neurodegenerative diseases, suggesting that neuronal population-specific therapeutic approaches may be warranted [204].

## 4. Conclusions and Future Research Directions

Current experimental studies on animal brains suggest that radiation induces aging in neural stem cells; mature and immature neurons; glial cells, including astrocytes, microglia, and oligodendrocytes; and endothelial cells of cerebral vessels. Cumulatively, these effects result in brain aging, leading to cognitive impairment and the development of aging-related brain disorders in individuals who are exposed to radiation, such as survivors from the Chernobyl nuclear power plant accident or individuals receiving radiotherapy. At the molecular level, radiation-induced oxidative stress and neuroinflammation may trigger different signal transduction pathways, resulting in the shortening of telomeres in brain cells, and eventually, brain aging.

Our current understanding of radiation-induced brain aging remains quite limited. With an increase in deep-space exploration, including space tourism, and the use of IR in medical diagnosis and treatment, extensive studies are required to obtain an in-depth understanding of how low-dose radiation affects brain aging and the molecular mechanisms that underlie this process. Furthermore, most radiation-related brain aging studies involve high doses of radiation, and few studies have examined the sensitivity of each CNS cell type and its progenitors to radiation and radiation-induced aging.

Senescent cells are considered effective treatment targets because they accumulate due to aging and other exogenous effects. Senotherapeutics drugs, a new class of drugs, can selectively kill senescent cells (senolytics) or suppress their disease-causing phenotypes (senomorphics/senostatics). Since 2015, several senolytics have been identified and examined via clinical trials. Preclinical data indicate that senolytics alleviate disease related effects in numerous organs, improve physical function and resilience, and suppress all causes of mortality, even among old patients [209]. In addition, new drugs can delay the patient’s disease recurrence. Accurate assessment of radiation response may provide the possibility to increase the sensitivity of cancer cells to radiation therapy while reducing damage to normal tissues [202].

Many senolytics have already been shown to be effective as they mediate the activation or inactivation of redox-sensitive hubs. Consequently, ROS-dependent pathways that specifically mediate the apoptosis of senescent cells may represent novel preventive/therapeutic targets for increasing treatment efficacy. As cells divide, telomere shortening, a process linked to cellular senescence occurs. Therefore, although senolytics temporarily alleviate cellular senescence and its deleterious effects, they could potentially cause accelerated aging and related dysfunction [210]. Drugs targeting aging-related mitochondrial dysfunction or specifically targeting mitochondrial ROS, may also allow alterations in the SASP and downstream negative outcomes. However, further elucidation of the complex mechanisms via which redox-regulated signaling pathways or mitochondria affect the SASP are required. It is necessary to note that exacerbated antioxidation could also lead to severe adverse effects. Only a tight control of redox homeostasis can eventually allow effective senomorphic-based therapies [211].

Hence, we propose that future exploration should focus on the following areas: (1) effect of low-dose IR on the aging of different cell types; (2) radiosensitivity of different progenitors and differentiated cells in the brain to radiation-induced aging; (3) epidemiology of brain aging in patients exposed to frequent radiodiagnosis and radiotherapy for brain disorders; and (4) the application of different -omics approaches for understanding the molecular mechanisms underlying low dose radiation-induced brain aging. This information could significantly aid in the development of protective and therapeutic approaches against radiation-induced brain aging and other related neurological and neuropsychological disorders.

## Figures and Tables

**Figure 1 cells-10-03570-f001:**
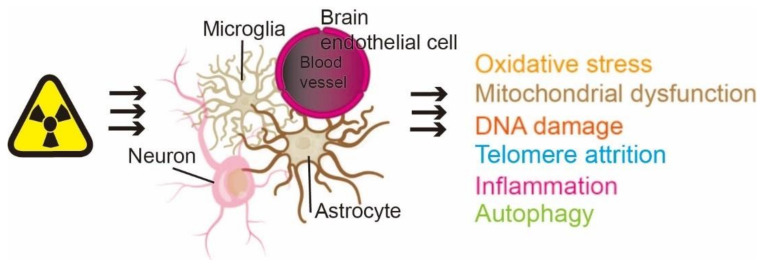
Radiation-induced brain aging includes oxidative stress, mitochondrial dysfunction, DNA damage, telomere attrition, inflammation, and autophagy.

**Figure 2 cells-10-03570-f002:**
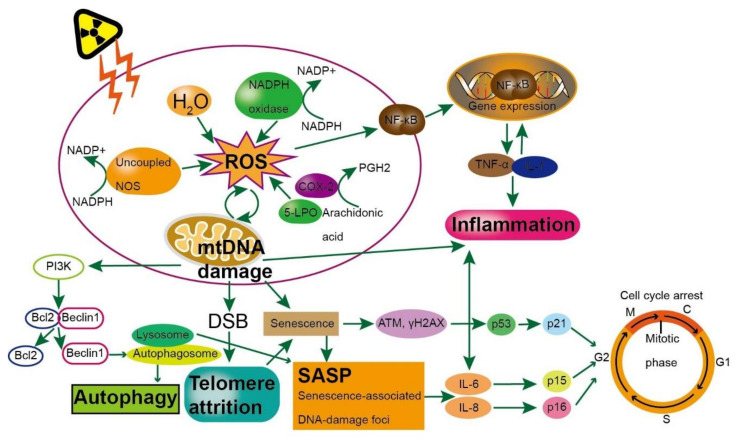
Interactions between various mechanisms of brain aging induced by ionizing radiation.

**Table 1 cells-10-03570-t001:** Medical conditions caused by different sources of low-dose/dose rate IR.

Radiation Source	Effects	References
Medical radiation (radiographs, computed tomography scans)	Cardiovascular disease, premature aging, inflammation, and neurodegenerative diseases	[16,17,21]
Natural background radiation	Inflammation, immunosenescence, thyroid cancer, and childhood leukemia	[41,42] [43,44]
Nuclear disasters	“Chernobyl AIDS,” CNS damage, premature aging, atherosclerosis, and senile encephalopathy	[28,29,30] [31,32]

**Table 2 cells-10-03570-t002:** Radiation-induced senescence in different cell types.

Cell Types	Models	Radiation Type & Dose/Dose-Rate	Radiation-Induced Changes	Reference
Microglia	Murine microglial cells BV2 and neuronal cells HT22	3 Gy/min (Clinac iX) (X-ray) 2 Gy/min (X-ray irradiator)	SA-β-Gal, p16^INK4a^, MMP3↑	[47]
	Primary microglia from adult male C57BI6/J mice	Single dose of 10/20 Gy at a dose rate of 3 Gy/min (Clinac iX) (X-ray)	SA-β-Gal, p16^INK4a^↑	[47]
Astrocytes	Non-cancerous tissue from cancer patients having received cranial radiation	IR (X-Rad 320 biologic irradiator) (X-ray)	p16^INK4a^, Hp1γ↑	[48]
	Primary human astrocytes	0.5–20 Gy (X-ray)	SA-β-Gal, p16^INK4a^, p21, IL-1, IL-6, IL-8↑IGF-1, GFAP↓ DNA damage	[48]
Brain endothelial cells	ATCC-derived murine brain endothelial cells, bEnd.3	X-ray (20 Gy)	SA-β-Gal, p21, p16^INK4a^, ICAM-1, PAI-1↑	[49]
Neurons	Male rats aged 8, 18 or 28 months	Whole-brain radiation with a single dose of 10 Gy (X-ray)	Greater inflammatory response; decrease in newborn neurons	[50]

## Data Availability

Data sharing is not applicable to this article, as no datasets were generated or analyzed during the present study.

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
