# Peer review of "Ionizing Radiation-Induced Brain Cell Aging and the Potential Underlying Molecular Mechanisms"

_cells, 2021, doi:10.3390/cells10123570_

Round 1

Reviewer 1 Report

The manuscript under review by Wang et al. overviews what is known to date about effects of ionizing radiation (IR) on brain cells and how these effects resemble and promote aging. The manuscript is well-structured and starts from describing how different types of brain cells are affected by IR and further explores IR-induced effects such as oxidative stress, mitochondrial dysfunction, telomere attrition, DNA damage, and inflammation. Some changes however can be made to make it better:

  1. “3. Mechanisms of radiation-induced brain aging” rather effect than mechanisms or at least in this context it is hard to distinguish them. It would be beneficial to add in the beginning description of direct physical mechanisms of IR-induced damage such as DNA damage and production of radicals and further proceed to description of further biological effects.
  2. Figure 2 summarizes IR-induced effects in the brain. It is not quite clear why the figure caption refers to the scheme as “protocol”. Also the connections between the different effects are not clear. For example, oxidative stress is shown to be connected with mitochondrial dysfunction and telomere attrition, while it also plays a significant role in DNA damage and inflammation. Similarly, mtDNA damage has been recently implicated in inflammation and immune response. Generally, all the effects are interconnected and it can be indicated. Also the scheme would benefit from inclusion of direct physical mechanisms of IR-induced damage.
  3. “3.1. Oxidative stress” would benefit from clearer description of IR-induced ROS/RNS generation as it is an important damaging factor. The authors mention several contributing factors, but some parts of description may be confusing. For example, the sentence in the third paragraphs “ROS is a generated owing to the participation of neutrophils and macrophages in inflammation and mitochondrial respiration” is confusing. ROS are generally generated during inflammatory response in immune cells, also ROS are independently generated during mitochondrial respiration by ETC (electron transfer chain) in all cell types and if we are talking about ionizing radiation – ROS are produced by radiolysis of water. These are three independent sources that can independently contribute to ROS production and consequent oxidative stress during IR.

Author Response

We thank you for all your constructive comments and suggestions on our manuscript. The point-to-point response based on your suggestion is as follows:

  • “3. Mechanisms of radiation-induced brain aging” rather effect than mechanisms or at least in this context it is hard to distinguish them. It would be beneficial to add in the beginning description of direct physical mechanisms of IR-induced damage such as DNA damage and production of radicals and further proceed to description of further biological effects.

Reply: We have described the direct physical mechanism of IR induction as you suggested. Please check Line 314-367.

“Mechanisms” has been changed to “effect”. Please check Line 306.

  • Figure 2 summarizes IR-induced effects in the brain. It is not quite clear why the figure caption refers to the scheme as “protocol”. Also the connections between the different effects are not clear. For example, oxidative stress is shown to be connected with mitochondrial dysfunction and telomere attrition, while it also plays a significant role in DNA damage and inflammation. Similarly, mtDNA damage has been recently implicated in inflammation and immune response. Generally, all the effects are interconnected and it can be indicated. Also the scheme would benefit from inclusion of direct physical mechanisms of IR-induced damage.

Reply: Figure 2 has been revised according to your suggestion. Please check Figure 2. The “protocol” in Figure note has been revised, please check Line 370-371.

  • “3.1. Oxidative stress” would benefit from clearer description of IR-induced ROS/RNS generation as it is an important damaging factor. The authors mention several contributing factors, but some parts of description may be confusing. For example, the sentence in the third paragraphs “ROS is a generated owing to the participation of neutrophils and macrophages in inflammation and mitochondrial respiration” is confusing. ROS are generally generated during inflammatory response in immune cells, also ROS are independently generated during mitochondrial respiration by ETC (electron transfer chain) in all cell types and if we are talking about ionizing radiation – ROS are produced by radiolysis of water. These are three independent sources that can independently contribute to ROS production and consequent oxidative stress during IR.

Reply: We have described the generation of IR-induced ROS/RNS in more detail according to your suggestion. Please check Line 377-390.

As for the confused statement you mentioned, we have deleted it under our consideration. Please check Line 409-411.

Reviewer 2 Report

Authors compiled evidence of the effects of ionizing radiation (IR) in the brain. The manuscript is readable; however, important sections of the text seem to lose the focus of the review.

Section 1. Introduction

Line 40. “With aging, there is an accumulation of changes in chromosomes, nucleic acids, proteins, and other macromolecules [2].” It is too early to talk about molecular aspects of aging in the first paragraph.

Line 63. The connection from brain aging to IR is weak. Among many other senescence-inducing aging in the brain, why authors decide to focus on IR?

Line 83. Talking about lymphocytes seems to be out of context in the overall goal of the review.

Section 2.

The compilation of evidence of IR-induced aging in different brain cell types is satisfactory.

Section 3.

This entire section loses the focus of the review. It is too long and evidence of the listed mechanisms induced by IR in brain cell is very limited. Overall, there are parts in the text that are redundant and some paragraphs should be shortened or are unnecessary. Below are some examples:

Line 313. “Live animal studies have demonstrated that dietary antioxidant supplements can attenuate the effects of IR [93].” Are these effects related to senescence-associated phenotype in brain cells?

In the subsection 3.3, there is a detailed description of the shelterin complex and telomeres. Do authors think that it is necessary for the overall purpose of the review? From lines 375 to 445 (four paragraphs), there is no evidence of the effect of IR in telomere attrition. Authors may make this subsection shorter and add evidence of the effects of IR in telomere attrition in the brain.

Line 446. “IR can also alter the function of lymphocytes via oxidative damage.” Why adding studies in lymphocytes?

Lines 455-471. It is not clear what tissues are affected in the studies referenced in this paragraph. Why authors include studies in fibroblasts?

Line 481. “The reduction in telomere length in microglia may reduce the ability of these cells to respond appropriately to CNS damage or infection, leading to apoptosis [158].” This is the only reference about the effect of IR in telomere attrition in a brain cell type.

In the subsection 3.4, from eight paragraphs, only the lines 572-575 refer to IR-induced DNA damage in brain cells.   

Autophagy participates in eliminating damaged mitochondria and is involved in DNA repair and inflammation. Authors may consider including a subsection about the effects of IR in autophagy in brain cells.

A section about potential therapies to prevent brain aging caused by IR is missing. For example, microglia depletion or senolytic drugs.

Author Response

We thank you for all your constructive comments and suggestions on our manuscript. The point-to-point response based on your suggestion is as follows:

Section 1. Introduction

Line 40. “With aging, there is an accumulation of changes in chromosomes, nucleic acids, proteins, and other macromolecules [2].” It is too early to talk about molecular aspects of aging in the first paragraph.

Reply: This sentence has been deleted.

Line 63. The connection from brain aging to IR is weak. Among many other senescence-inducing aging in the brain, why authors decide to focus on IR?

Reply: Thank you very much for your advice. We have added evidence on the link between brain aging and radiation exposure. Please check Line 63-72.

Line 83. Talking about lymphocytes seems to be out of context in the overall goal of the review.

Reply: This sentence has been deleted. Please check Line 103.

Section 2.

The compilation of evidence of IR-induced aging in different brain cell types is satisfactory.

Reply: Thank you very much for your approval of Section 2, which is a great encouragement to us. Best regards.

Section 3.

This entire section loses the focus of the review. It is too long and evidence of the listed mechanisms induced by IR in brain cell is very limited. Overall, there are parts in the text that are redundant and some paragraphs should be shortened or are unnecessary. Below are some examples:

Line 313. “Line 313. “Live animal studies have demonstrated that dietary antioxidant supplements can attenuate the effects of IR [93].” Are these effects related to senescence-associated phenotype in brain cells?

Reply: After consideration, we find that this sentence does not relate to senescence-associated phenotype in brain cells, so we delete this sentence. Please check Line 418-420.

In the subsection 3.3, there is a detailed description of the shelterin complex and telomeres. Do authors think that it is necessary for the overall purpose of the review? From lines 375 to 445 (four paragraphs), there is no evidence of the effect of IR in telomere attrition. Authors may make this subsection shorter and add evidence of the effects of IR in telomere attrition in the brain.

Reply:  The change has been done according to your suggestion. Please refer to Lines 480-594.

Line 446. “IR can also alter the function of lymphocytes via oxidative damage.” Why adding studies in lymphocytes?

Reply: After consideration, we think this sentence is redundant, so we delete it. Please check Line 550-552.

Lines 455-471. It is not clear what tissues are affected in the studies referenced in this paragraph. Why authors include studies in fibroblasts?

Reply: After consideration, we think this part of the description is not necessary, so we delete it. Please check Line 578-594.

Line 481. “The reduction in telomere length in microglia may reduce the ability of these cells to respond appropriately to CNS damage or infection, leading to apoptosis [158].” This is the only reference about the effect of IR in telomere attrition in a brain cell type.

Reply: Some evidence of the effects of IR in telomere attrition in the brain have been added, please check Line 559-577.

In the subsection 3.4, from eight paragraphs, only the lines 572-575 refer to IR-induced DNA damage in brain cells.

Reply: Some evidence of IR-induced DNA damage in brain cells have been added, please check Line 627-634.

Autophagy participates in eliminating damaged mitochondria and is involved in DNA repair and inflammation. Authors may consider including a subsection about the effects of IR in autophagy in brain cells.

Reply: Thank you very much for your advice. We have added a section on autophagy. Please check Line 750-788.

A section about potential therapies to prevent brain aging caused by IR is missing. For example, microglia depletion or senolytic drugs.

Reply: We've added a section about potential therapies to prevent brain aging caused by IR is missing on your suggestions. Please check Line 806-829.

The revised manuscript has been uploaded, please check the attachment.

Reviewer 3 Report

Overall Comments:

Authors in this manuscript thoroughly reviewed the radiation implicated brain aging and its underlying molecular mechanisms. In the first part, the authors provided backgrounds and needs for this study: both the worldwide population and the aged people sharply increase and the three cases of ionizing radiation exposure is able to accelerate aging processes. Next, they explain the influence of radiation on the cellular senescence of different types of cells in brain. Finally, they summarize the underlying mechanisms of radiation implicated brain aging and degenerative diseases including Alzheimer’s disease. This review manuscript is well organized and presents clear information for the subject. However, they do not include the most recent information (since early 2021) and the figures in the manuscript do not present clear and enough information. In addition, they may need more information on the effects of Natural background radiation on the brain aging because the everyday radiation may be the more general factor affecting people. If the authors revise these issues, this manuscript is good enough to be considered for publication in “Cells”.

Comments:

  1. ‘Figure 1’ is not necessary because the population shift is already recognized by most people and has been presented in other review papers.
  2. ‘Figure 2’ presents “five radiation-induced brain aging features” with brief explanation about the underlying mechanisms. We suggest that the authors present the five brain aging features with more information in Figure 2 and provide an additional figure describing the underlying mechanisms more detailed. If the number of figures is limited, authors may present two figures without “Figure 1”.
  3. If available, we suggest that the author present the figure describing their overall ideas about radiation implicated in brain aging.
  4. Compared to “Medical radiation” and “Nuclear disasters”, the influence of “Natural background radiation” on aging is less described in the manuscript. We suggest that the authors give more information on the everyday exposure to the radiation and its effects and deal with the safety and the risk of both “Medical radiation” and “Natural background radiation” in disease development and aging.
  5. The authors did not include the most recent information (since 2021.01) except one reference. We suggest that they review and add the most recent journals as reference if additional information is reported.
  6. Line 57~62: The authors need to add references to what is written in the introduction. (https://doi.org/10.3390/ijms22073342, https://doi.org/10.3390/jcm10132773). These papers include the information related to brain aging.
  7. Line 191~198: Previous studies has been reported that microglia and astrocytes interact with each other, and astrocytes are known to participate in the immune system. We suggest that it will be better if you add this content in the manuscript. (10.3389/fimmu.2019.01314)
  8. The authors should improve the resolution of each figure.

Author Response

We thank you for all your constructive comments and suggestions on our manuscript. The point-to-point response based on your suggestion is as follows:

(1). ‘Figure 1’ is not necessary because the population shift is already recognized by most people and has been presented in other review papers.

Reply: “Figure 1” has been deleted.

(2). ‘Figure 2’ presents “five radiation-induced brain aging features” with brief explanation about the underlying mechanisms. We suggest that the authors present the five brain aging features with more information in Figure 2 and provide an additional figure describing the underlying mechanisms more detailed. If the number of figures is limited, authors may present two figures without “Figure 1”.

Reply: Changes have been made according to your suggestion. Please check Figure 1 and Figure 2.

(3). If available, we suggest that the author present the figure describing their overall ideas about radiation implicated in brain aging.

Reply: Changes have been made according to your suggestion. Please check Figure 2.

(4). Compared to “Medical radiation” and “Nuclear disasters”, the influence of “Natural background radiation” on aging is less described in the manuscript. We suggest that the authors give more information on the everyday exposure to the radiation and its effects and deal with the safety and the risk of both “Medical radiation” and “Natural background radiation” in disease development and aging.

Reply: Thank you very much for your advice. We have added description about natural background radiation and dealt with the safety and the risk of both “Medical radiation” and “Natural background radiation” in disease development and aging. Please check Line 84-92, 96-99, 102-122.

(5). The authors did not include the most recent information (since 2021.01) except one reference. We suggest that they review and add the most recent journals as reference if additional information is reported.

Reply: Changes have been made according to your suggestion. Please check reference 8, 9, 27, 39, 92, 209, 210, 211.

(6). Line 57~62: The authors need to add references to what is written in the introduction. (https://doi.org/10.3390/ijms22073342, https://doi.org/10.3390/jcm10132773). These papers include the information related to brain aging.

Reply: Changes have been made according to your suggestion. Please check reference 8,9.

(7). Line 191~198: Previous studies has been reported that microglia and astrocytes interact with each other, and astrocytes are known to participate in the immune system. We suggest that it will be better if you add this content in the manuscript. (10.3389/fimmu.2019.01314).

Reply: Changes have been made according to your suggestion. Please check Line 223-225 and reference 67.

(8). The authors should improve the resolution of each figure.

Reply: All figures have a resolution of 300dpi.

The revised manuscript has been uploaded, please check the attachment.

Round 2

Reviewer 2 Report

Authors addressed all suggestions and made significant editing in the review, which is now more comprehensive for the selected topic.

Reviewer 3 Report

The authors faithfully answered the reviewer's questions and corrected the issues as suggested. I think the revised manuscript entitled “Ionizing radiation-induced brain cell aging and the potential underlying molecular mechanisms.” is suitable for publication in “Cells”.